# Methodological considerations for assessing elder mistreatment of older adults with cognitive impairment: A scoping review protocol

**Karen E. Schlag**[1]*, **Rebecca Czyz**[2], **Monique R. Pappadis**[1,3]

**1** Sealy Center on Aging, The University of Texas Medical Branch, Galveston, Texas, United States of America, **2** School of Medicine, The University of Texas Medical Branch, Galveston, Texas, United States of America, **3** Department of Population Health and Health Disparities, School of Public and Population Health, The University of Texas Medical Branch, Galveston, Texas, United States of America

* keschlag@utmb.edu

## Abstract

Elder mistreatment (EM) of older persons with cognitive impairment is thought to be grossly underestimated in part due to communication barriers experienced by victims and a lack of consistent screening and reporting, which can skew current understandings of this problem. To improve EM risk and prevalence screening in relation to cognitive impairment, it is important to understand specific approaches for implementing assessment tools and interventions for members of this population. Accordingly, this scoping review (OSF registration osf.io/759k3) will identify, summarize, and compare methodological considerations adopted in studies assessing EM risk and occurrence among older persons with varying degrees of cognitive impairment. Through mapping out existing strategies and approaches used to develop, test, and implement EM screening tools or interventions, this review will outline previously identified recommendations and challenges pertinent to future EM assessment, reduction, and prevention efforts. We will follow the Preferred Reporting Items for Systematic Reviews and Meta-Analyses extension for Scoping Reviews (PRISMA-ScR) guidelines and apply the Arksey and O'Malley (2005) scoping review framework. We will identify relevant studies by comprehensively searching electronic databases, including Ovid (Medline), Cumulative Index of Nursing and Allied Health Literature (CINAHL), Cochrane Trials (CENTRAL), and Elsevier's Scopus. Reference lists of included studies will also be examined. For article selection, we will use Covidence software to guide a two-step process of title/abstract and full article screening, which will allow us to identify eligible studies based on our inclusion and exclusion criteria that follows the Study Design, Data, Methods, Outcomes (SDMO) framework. A standardized data extraction tool will be used to collect information related to authors, year of publication, research objectives, sample and study design characteristics, measures, analysis, outcomes, limitations, and study conclusions and implications related to cognition. Data will be analyzed using a thematic approach and presented through the reporting of descriptive statistics and summaries.

**Data availability statement:** No datasets were generated or analysed during the current study. All relevant data from this study will be made available upon study completion.

**Funding:** This study was supported by a grant awarded to Dr. Monique R. Pappadis from the National Institute on Aging, AG078519, https://www.nia.nih.gov/. Funders had no role in the study design, data collection and analysis, decision to publish or preparation of the manuscript.

**Competing interests:** The authors have declared that no competing interests exist.

## Introduction

Elder mistreatment (EM) refers to actions or nonactions that cause harm or risk of harm to older adults [1], and is characterized by various types of abuse, including physical, emotional, sexual, and financial abuse, as well as neglect [2]. Victims of EM are at greater risk of experiencing adverse health outcomes over time [3] and increased medical costs [4]. It is estimated that one in six adults aged 60 years and older worldwide have experienced some form of EM [1]. Common risk factors associated with older adults who experience EM include an increased need for physical assistance, lower cognitive functioning or other mental health issues, and reduced social support [2,5].

Older adults with cognitive impairment, including Alzheimer's Disease and Related Dementias (ADRD), experience a higher risk and incidence of abuse compared to those with healthy cognitive functioning [6–8]. Behavioral symptoms associated with dementia, such as agitation and aggression, as well as an increased likelihood of physical impairment, other illnesses, or social isolation, have been linked to a heightened risk of EM in older adults with cognitive impairment, especially as cognitive functioning declines [9–11]. The occurrence and severity of different types of abuse may also vary according to the degree of cognitive impairment [12,13]. While abuse against older adults with impaired cognitive functioning can occur by professionals, such as home care workers [14] or nursing home staff [7], EM is often perpetrated by family members [15]. Research has identified several caregiver-related risk factors for EM in older adults with cognitive impairment, including caregiver anxiety, depression, lack of social support, extended caregiving hours, and substance abuse [6,9,16–19].

EM among older adults with cognitive impairment is thought to be grossly underestimated, partly because broad abuse estimates often rely on reports from older adults with healthy cognitive functioning who are willing to share their experiences [20]. Individuals with conditions such as mild cognitive impairment (MCI) or ADRD may be more reticent or unable to report occurrences of mistreatment [21]. Furthermore, older adults with impaired cognitive functioning who are highly dependent on regular care may be especially reluctant to speak out against family members who provide necessary support [22–24]. Additionally, healthcare providers do not routinely screen for or report mistreatment of older patients with cognitive impairment [23], which may be due to factors such as time constraints or differing perspectives on what constitutes abuse [25,26].

Reported prevalence of EM for older adults with cognitive impairment has also varied across studies. Differences in methodological approaches and types of abuse studied have contributed to a lack of consistency and comparability in EM findings, both among general populations of older adults and those with cognitive impairment [12,27]. For instance, reported abuse rates for older adults with ADRD are typically higher in studies that use instruments with established psychometric properties and rely on family caregiver informants, compared to studies that use not yet validated measures or depend on care recipients or professionals to complete the assessments [12]. Studies with longer assessment periods (e.g., one year compared to three months) have also reported higher abuse rates for people with dementia [12]. Moreover, screening instruments used to identify EM exposure across different populations are not frequently validated across cultural groups and may not fully address the needs of vulnerable older adults, such as those with ADRD [28,29]. The underreporting of EM among older persons with cognitive impairment, along with inconsistency in research findings on abuse prevalence in this population, can impact understanding of the unique EM risk characteristics for these individuals.

While several validated screening tools have been developed worldwide to assess direct mistreatment or characteristics of abuse vulnerability for older adults, none have been specifically designed to focus exclusively on older persons with impaired cognitive functioning. EM

interventions targeting family caregivers of older adults with ADRD have been developed to provide educational materials and support resources to families in relation to caregiving and stress management [30,31], although they have not been tested for their effectiveness in reducing EM or related risks on a large scale [32]. Observational and intervention research designs typically discern EM exposure and characteristics of abuse vulnerability through population surveys, wherein older adults, family caregivers, or health professionals complete survey measures on the perceived occurrence and frequency of specific abuse types [29,33]. Moreover, while a prior review has conveyed how variation in EM definitions and measurements impact EM findings for older adults with dementia compared to those without [12], less is known about the practical considerations and modifications adopted by researchers when developing and implementing EM assessments and intervention strategies for individuals with varying levels of cognitive impairment, many of whom may be unable to fully communicate their experiences of abuse.

The purpose of this proposed scoping review is to identify, summarize, and compare the unique methodological considerations and practical approaches that have been used to develop and implement EM screening tools or interventions for individuals with cognitive impairment or their family members. A scoping review offers a method for providing an overview of the literature on a topic of choice, including reflecting on characteristics across different studies [34]. In using this approach to map out existing literature on EM assessment and intervention strategies for persons with varying levels of cognitive impairment, this review will inform future studies by summarizing previously identified challenges and recommendations for implementing EM screenings and interventions to people with impaired cognition and their caregivers. This review will also identify reported gaps for future research. Finally, findings are expected to benefit healthcare practitioners by highlighting practical aspects to consider when evaluating potential abuse in older patients with cognitive impairment.

## Materials and methods

This scoping review protocol has been registered with OSF (osf.io/759k3) and follows reporting guidelines provided by the Preferred Reporting Items for Systematic Reviews and Meta-Analyses extension for Scoping Reviews (PRISMA-ScR) Checklist [35]. To conduct this review, we will also follow the framework proposed by Arksey and O'Malley [36] for conducting scoping reviews, which includes the following stages: 1) identifying the research question; 2) identifying relevant studies; 3) selecting studies; 4) charting the data; 5) collating, summarizing and reporting results.

### Identifying the research question

To provide an overview of the methodological considerations adopted in research on EM risk and occurrence among older persons with cognitive impairment, this study will be guided by the following research question: What methodological considerations have researchers taken into account for studies assessing EM risk and occurrence among older adults with varying degrees of cognitive impairment? To further explore this question, this review will also address the following sub-questions:

1. For research developing or applying EM risk or occurrence screenings, what design modifications or considerations have been made to include older adults with varying degrees of cognitive impairment?

2. For intervention development and testing, what design modifications or considerations have been made to include older adults with varying levels of cognitive impairment?

## Identifying relevant studies

To identify relevant studies for this review, a medical librarian will conduct comprehensive searches in the following electronic databases: Ovid (Medline), Cumulative Index of Nursing and Allied Health Literature (CINAHL), Cochrane Trials (CENTRAL), and Elsevier's Scopus. These databases that were selected based on availability of the resources and relevancy of the content covered. Medline is the premier biomedical database, CINAHL covers other health sciences fields, Cochrane Trials provides evidence-based literature, and Scopus provides a large and multidisciplinary scope of the scholarly literature. Reference lists of included studies will also be reviewed to identify any relevant research that may have been missed during the initial database searches.

To provide an overview of the methodological considerations that have been adopted in studies assessing EM risk and occurrence among older adults with varying degrees of cognitive impairment, this review will employ a search strategy that includes a combination of MeSH and keyword terminology on the topic. The search terms will consist of the following: ((Alzheimer OR dementia OR cognition OR cognitive) AND (elder abuse OR (domestic abuse OR physical abuse OR intimate partner violence OR emotional abuse OR self-neglect OR sex offenses OR bullying AND Aged))). Various synonyms of the included terms will be utilized, along with proximity searching. (Fig 1). This search strategy was piloted to ensure the appropriateness of the keywords and the relevance of the results produced and initiated April 25, 2024. Moving forward, search results will be extracted and imported into Covidence, a web-based tool used to streamline the process of knowledge synthesis for reviews, and all duplicates will be removed.

1. exp Alzheimer Disease/ or exp Dementia/ or (alzheimer* or dementia* or amentia*).mp.

2. Cognition Disorders/ or Cognitive Dysfunction/ or Cognitive Aging/ or exp Cognition/ or (cognition* or cognitive* or "mental deterioration*" or "mental decline*").mp.

3. 1 or 2

4. exp Elder Abuse/ or ((aged or elder* or senior* or "old* adult*" or "old* person*" or "old* people*" or "old* individual*" or gerontol* or geriatric* or ageing or aging or retir* or octogenarian* or nonagenarian* or septuagenarian* or sexagenarian* or "old* man*" or "old* woman*" or "old* men*" or "old* women*" or "old* minorit*" or senescent or senile* or "later life*") adj2 (abus* or maltreat* or neglect* or mistreat* or exploitat* or restrain* or illtreat* or "ill treat*" or hit* or victimi* or manhandl* or "socially isolate*" or exclude or exploit* or bully* or violen* or self-neglect* or intimidat* or threaten* or abandon* or fraud* or rape* or scam* or crime* or extort* or hoax or swindle* or coerce* or assault* or harm or attack* or beat* or aggressi* or duress or "inadequate care" or "involuntary treat*")).mp.

5. ((Domestic Violence/ or Physical Abuse/ or exp Intimate Partner Violence/ or Emotional Abuse/ or exp Self-Neglect/ or Sex Offenses/ or exp Bullying/) and exp Aged/) or ((aged or elder* or senior* or "old* adult*" or "old* person*" or "old* people*" or "old* individual*" or gerontol* or geriatric* or ageing or aging or retir* or octogenarian* or nonagenarian* or septuagenarian* or sexagenarian* or "old* man*" or "old* woman*" or "old* men*" or "old* women*" or "old* minorit*" or senescent or senile* or "later life*") adj2 ("domestic violence*" or "domestic abuse*" or "family violence*" or "family abuse*" or "interpersonal violence*" or "structural violence*" or "intimate partner abuse*" or "intimate partner violence*" or "dating violence*" or "emotional abuse*" or "psychological abuse*" or "sexual abuse*" or "sexual violence*" or "sexual assault*" or "sex offense*" or "patient abuse*" or "verbal abuse*" or "financial abuse*" or "financial exploit*")).mp.

6. 4 or 5

7. 3 and 6

**Fig 1. Scoping review search example using Medline.** Note: This example search using Medline was run on 3/6/2024 and produced 950 results.

## Study selection

This review will use the Study Design, Data, Methods, Outcomes (SDMO) framework [37] to guide title and abstract and full-text screening of studies. Inclusion and exclusion criteria will be refined iteratively, comparing screened abstracts to the established criteria to ensure relevance to the scope of the inquiry.

Three independent reviewers will screen all eligible studies for abstract/title and full-text review. Results of these screenings will be discussed regularly throughout this process, and updates to the final data-charting form will be made as needed. Any conflicts will be reviewed and discussed until consensus is reached. In the case of missing data, efforts will be made to contact the original study investigators to obtain unreported data or relevant information.

**Inclusion criteria.** When determining inclusion criteria, we will draw from the SDMO framework for reviewing research methodologies. This framework has been recommended for methodologically focused reviews, as it guides identification of study design, data, and outcome elements relative to the research question [37]. Inclusion criteria for the present study will be based on the following components of peer-reviewed articles: study types, data types, methods types, and outcome types.

1. Types of studies: Studies included in this review should be published in English and incorporate a method for assessing EM or the risk of EM. This includes studies designed to develop or test a tool or intervention. Eligible study designs include randomized control trials (RCTs), quasi RCTs, and observational studies (including cohort and case control studies). Cross-sectional and longitudinal studies will be considered as will quantitative data from mixed methods studies.

2. Types of data: Studies that use self-report or observational data will be included.

3. Types of methods: Studies that examine EM in clinical settings, home environments, or service facilities in high, middle and low-income countries will be included. Participants may consist of older adults with cognitive impairment of varying severity who experience EM, adult family caregivers of these older adults, or healthcare providers, including professional caregivers. Studies will be included if they describe and use a measurement tool specifically designed to identify risk factors, signs, or symptoms of elder mistreatment.

4. Types of outcomes: Studies will be included if they report signs or symptoms of EM. EM includes but is not limited to any intentional or unintentional act of physical, psychological, sexual, and financial abuse, as well as neglect and abandonment. Accordingly, we will include studies that indicate the presence, reduction or lack of EM as primary outcomes.

We will also include studies having self-report health measures (e.g., well-being, depression) or quality of life measures as secondary outcomes for older adults with cognitive impairment of varying severity, or for family caregivers providing dementia-related care support.

**Exclusion criteria.**

1. Studies that are not published in English or that discuss EM screening tools or interventions through qualitative assessments, reviews, or grey literature (e.g., dissertations, conference abstracts) will be excluded.

2. Studies that do not incorporate use of an assessment tool to determine presence of EM or EM risk.

3. Studies focused on child or animal abuse, or on interpersonal violence in contexts other than elder abuse, will be excluded.

## Charting the data

Covidence software will be used to extract data on relevant methodological characteristics from each included peer-reviewed article that comprise overall approaches used to develop, test, or implement EM screenings or interventions for older adults in the context of cognitive impairment. A standardized data extraction form will be developed to capture characteristics including: authors, year of publication, research objectives, sample characteristics (e.g., population, sample size/demographics/response rate), study design characteristics (e.g., locale/ language, recruitment strategies, research design), study measures (e.g., application of elder mistreatment assessment, cognitive status for potential victims of EM, health status of older adults with cognitive impairment or family caregivers), quality assessment of study measures, methods of analysis, outcomes, limitations, and authors' conclusions or noted implications for considering cognition and making research design modifications for this population. The extraction data will be made publicly available in OSF.

## Summarizing, collating, and reporting the results

A narrative report will be used to summarize the extracted data in line with the following outcomes: study region, targeted participants, study design characteristics (including method of EM assessment and any special considerations), types of abuse and their prevalence, reported EM risk characteristics, and other health outcomes. Results will be presented in relation to the research questions and the broader study objectives. In consideration that collected data from this review may yield unanticipated concepts or categories, findings will be reported using a thematic approach to allow for flexibility in reflecting the data. Additionally, outlining gap identification will highlight areas where data on EM assessment for older adults with cognitive impairment is insufficient.

# Discussion

Findings from this study will provide an overview of the unique and practical considerations adopted by researchers when assessing EM and associated risk factors for older adults with varying levels of cognitive impairment. By summarizing effective tactics, challenges, and research gaps that have been previously reported in relation to EM assessment among this population, findings from this review will also inform efforts to develop EM screening tools and interventions specifically for these individuals. Moreover, the results from this study will have implications for healthcare practitioners by summarizing appropriate EM risk screening implementation strategies for older adults with impaired cognitive functioning.

One limitation of this review is its exclusion of studies conducted using qualitative methods and those not reported in English, which may result in overlooking relevant information about the design and implementation of EM assessments for older adults with cognitive impairment. Another limitation is that the reliability of the extracted data cannot be evaluated, as this scoping review will not assess the quality of the included studies.

Ultimately, this review will focus on studies that have intentionally incorporated and applied identifiable strategies for assessing for EM risk or occurrence as part of their methodology. By doing so, this review will map out previous practices used for implementing these tools and interventions in patient-care settings where individuals may have unique communication needs. Findings will be disseminated through submission to a scholarly journal in the fields of gerontology or public health. While amendments to the proposed scoping review are not anticipated, any issues and adjustments that arise during the implementation of this protocol will be documented in the methods section of the review paper.

## Supporting information

**S1 Fig. PRISMA 2020 flow diagram.**
(DOCX)

**S2 Fig. PRISMA-P checklist.**
(DOCX)

## Acknowledgements

The authors would like to acknowledge Alison Hansen, MLIS, for providing guidance and conducting searches of electronic databases and Amber McIlwain, MS, for editing and formatting the manuscript.

## Author contributions

**Conceptualization:** Karen E. Schlag, Rebecca Czyz, Monique R. Pappadis.

**Data curation:** Karen E. Schlag, Rebecca Czyz, Monique R. Pappadis.

**Formal analysis:** Karen E. Schlag, Rebecca Czyz, Monique R. Pappadis.

**Funding acquisition:** Monique R. Pappadis.

**Investigation:** Karen E. Schlag, Monique R. Pappadis.

**Methodology:** Karen E. Schlag, Monique R. Pappadis.

**Supervision:** Monique R. Pappadis.

**Visualization:** Karen E. Schlag, Monique R. Pappadis.

**Writing – original draft:** Karen E. Schlag, Rebecca Czyz.

**Writing – review & editing:** Karen E. Schlag, Monique R. Pappadis.

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
