## [Decision Letter · Decision Letter 0]

7 Jan 2025

Dear Dr. Schlag, 

Thank you for submitting your manuscript to PLOS ONE. After careful consideration, we feel that it has merit but does not fully meet PLOS ONE’s publication criteria as it currently stands. Therefore, we invite you to submit a revised version of the manuscript that addresses the points raised during the review process.

We look forward to receiving your revised manuscript.

Kind regards,

Patricia Anne Morris

Academic Editor

PLOS ONE

Journal Requirements:

2. Please note that funding information should not appear in the Acknowledgments section or other areas of your manuscript. We will only publish funding information present in the Funding Statement section of the online submission form. Please remove any funding-related text from the manuscript. 

4. Please include a separate caption for figure 1 in your manuscript.

**Additional Editor Comments:**

Thank you for the opportunity to review this protocol. I apologize for the significant delay in returning this to you. The process was delayed by reviewers who agreed to assess the work but were unable to complete the assignment in a timely manner. It is an important topic and the protocol is well conceived and well written. The reviewers for this protocol make a number of suggestions to improve clarity and to ensure that this protocol aligns with the goals of a scoping review. Please pay special attention to Reviewer 1's commentary on methodological characteristics, as more detail is required here, and to Reviewer 2's comments about the ability to make recommendations based on this review. These minor changes, along with some of the editorial suggestions from both reviewers, will ensure the protocol provides strong scaffolding for your final review. Thank you for your submission. 

Reviewers' comments:

Reviewer's Responses to Questions

**Comments to the Author**

1. Does the manuscript provide a valid rationale for the proposed study, with clearly identified and justified research questions?

Reviewer #1: Partly

Reviewer #2: Yes

2. Is the protocol technically sound and planned in a manner that will lead to a meaningful outcome and allow testing the stated hypotheses?

Reviewer #1: Yes

Reviewer #2: Yes

3. Is the methodology feasible and described in sufficient detail to allow the work to be replicable?

Reviewer #1: Yes

Reviewer #2: No

4. Have the authors described where all data underlying the findings will be made available when the study is complete?

Reviewer #1: Yes

Reviewer #2: Yes

5. Is the manuscript presented in an intelligible fashion and written in standard English?

Reviewer #1: Yes

Reviewer #2: Yes

You may also provide optional suggestions and comments to authors that they might find helpful in planning their study.

Reviewer #1: Thank you for the opportunity to review manuscript PONE-D-24-36659

“Methodological considerations for assessing elder mistreatment of older adults with cognitive impairment: A scoping review protocol.” Over the manuscript is well written and organized. The topic of elder mistreatment is important, particularly in the context of cognitive impairment, where the authors so clearly identified is not well understood. Research conducted in this area has the potential to make an important contribution to the literature and will be of interest to a wide audience.

Some suggestions are offered to add clarify and to strengthen the manuscript:

1. The protocol aims to map the “methodological characteristics and approaches” in studies that examine risk and occurrence of EM. The concept of methodological characteristics Page 10 describes sample and design characteristics. Are these the methodological characteristics of interest? The concept of methodological characteristics should be further developed.

2. Sub-question 1 identifies design modifications – is this different from characteristics? Also, modified from what? Will there be comparisons between studies that examine EM in older people in general and older people with cognitive impairment?

3. It is not clear if a similar scoping review/systematic review already exists on the topic. This should be identified.

4. Line 141 states that a quality appraisal will not be done. Given this is a scoping review, this is not necessary to state.

5. Given the research question/sub-questions identified, it is not clear how the findings from this review will identify the challenges in identifying the prevalence of EM (page 6 127-128).

6. Like the above comment, given this is a scoping review with no critical appraisal, it will not be possible to offer recommendations as stated in line 128.

7. Type of outcome – it is possible that a study could meet the inclusion criteria by reporting signs and symptoms of abuse but not identify a specific assessment tool. It is not clear if these studies would be included.

8. It is interesting that nothing has been discussed about the severity of cognitive impairment (e.g. MMSE) it only states “varying degrees”.

Reviewer #2: Thank you for the opportunity to review your manuscript. Your objective, justification, research questions, and inclusion criteria align nicely. Your topic is incredibly important and as a witness of various forms of elder mistreatment, I am so pleased to see your research plans!

My comments are very minor and are aimed to bring more clarity to your concepts to enhance rigor. Overall, I enjoyed this manuscript and found it very thorough and well done.

Line 65: you can put just “EM” instead of elder mistreatment

Line 127-128: “… this review will inform future research through identifying challenges and recommendations for developing and implementing EM screenings and interventions for individuals with cognitive impairment.” I suggest changing this wording – scoping reviews are not meant to identify recommendations, but they can map what recommendations have been made already.

Line 126, 188: Prevention strategies – these are the only two places prevention/mitigation strategies are mentioned. It seems to me this may not align well with your purpose/questions – perhaps reword or reconsider this for clarity.

Lines 155-158: Perhaps include why these databases were chosen.

Line 170: Covidence is fantastic for scoping reviews! However, your readers may not know what Covidence is. I suggest quickly adding something about Covidence being a software used to streamline the knowledge synthesis process

Line 172: Similar to the previous comment, what is the SDMO framework and why is using it important for the screening process where you are already following Arksey and O’Malley’s framework and using Covidence. How does SDMO come into play?

Line 185: It is nice to have very clear conceptual definitions in inclusion criteria for scoping reviews. Although you defined EM in the introduction, consider giving a clear definition of what you will consider as EM in your review within the inclusion criteria. For example, “EM includes but is not limited to any intentional or unintentional emotional, physical, financial, or sexual abuse or neglect……”

Line 190: What about any quantitative data within mixed-methods studies? Or quantitative data within reports from relevant associations or organizations?

Line 192: “types of methods” section. I wonder if including a definition here how you define ‘methodological considerations’ as written in your research questions will help your reader understand what about the studies and instruments you are interested in. I.e., methodological considerations included in this review are study designs, such as recruitment strategies, ……

Line 194: The word ‘older’ is written in purple, and is there a certain age you will consider as being an “older” adult?

Line 239: I am not sure not including qualitative studies is a limitation of this review as the purpose is to distinctly examine instruments.

Line 246-247: “highlight best practices for implementing these tools in patient-care settings with unique communication needs” this sounds as if you plan to draw conclusions about best practices for implementing tools. Scoping reviews cannot do this as they do not appraise evidence quality. Please reword this

It is obvious an immense amount of work has gone into this protocol. It is very well thought-out and intriguing. I honestly think your review will be great!! Thank you for this important work.

**Do you want your identity to be public for this peer review?** For information about this choice, including consent withdrawal, please see our Privacy Policy

Reviewer #1: **Yes: ** Rose McCloskey RN PhD

Reviewer #2: **Yes: ** Rachel MacLean

---

## [Author Response · Author response to Decision Letter 1]

20 Feb 2025

Thank you to the academic editor and the reviewers for the opportunity to revise our protocol manuscript. We have addressed each point below in red and have resubmitted our manuscript with the noted changes. These points have helped us to clarify strengthen areas in our paper.

Academic editor

Thank you for the opportunity to correct aspects with our manuscript style and file naming. We have made corrections on the title page and to some of the headings, renamed our Fig 1, included a caption in the manuscript (p. 8), and resaved this file as an image (.tif). We have also corrected the figure titles of supporting information (and their placement in the manuscript) along with their file naming.

2. Please note that funding information should not appear in the Acknowledgments section or other areas of your manuscript. We will only publish funding information present in the Funding Statement section of the online submission form. Please remove any funding-related text from the manuscript.

Thank you for bringing this to our attention. We have removed funding information from our title page and included it in the Funding Statement section of the online submission form.

The dataset that will support the findings of the scoping review will be created from the data collected and will be openly available in OSF. Our review was registered in OSF. (See line 127.)

4. Please include a separate caption for figure 1 in your manuscript.

Thank you for bringing this to our attention. We have made this correction on p. 8.

We reviewed the references to check that they followed the PLOS One style guidelines described on the links: https://journals.plos.org/plosone/s/submission-guidelines#loc-references and: https://www.nlm.nih.gov/bsd/uniform_requirements.html#electronic

We updated one reference [27] such that we replaced a protocol manuscript we had originally cited with the completed and published study.

Our original citation: Mohd Mydin FH, Mikton C, Choo WY, Shanmugam RH, Murray A, Yon Y, et al. PROTOCOL: Psychometric properties of instruments for measuring elder abuse and neglect in community and institutional settings: A systematic review. Campbell Syst Rev. 2023;19(3) :e1342. Epub 20230627. doi: 10.1002/cl2.1342. PubMed PMID: 37383829; PubMed Central PMCID: PMCPMC10296034.

New citation used in our revised paper: Mohd Mydin FH, Mikton C, Choo WY, Shunmugam RH, Murray A, Yon Y, Yunus RM, Hairi NN, Hairi FM, Beaulieu M, Phelan A. Psychometric properties of instruments for measuring abuse of older people in community and institutional settings: A systematic review. Campbell Syst Rev. 2024 Aug 29;20(3):e1419. doi: 10.1002/cl2.1419. PMID: 39211334; PMCID: PMC11358705.

Review Comments to the Author

Reviewer #1: Thank you for the opportunity to review manuscript PONE-D-24-36659

“Methodological considerations for assessing elder mistreatment of older adults with cognitive impairment: A scoping review protocol.” Over the manuscript is well written and organized. The topic of elder mistreatment is important, particularly in the context of cognitive impairment, where the authors so clearly identified is not well understood. Research conducted in this area has the potential to make an important contribution to the literature and will be of interest to a wide audience.

Thank you for your feedback!

Some suggestions are offered to add clarify and to strengthen the manuscript:

1. The protocol aims to map the “methodological characteristics and approaches” in studies that examine risk and occurrence of EM. The concept of methodological characteristics Page 10 describes sample and design characteristics. Are these the methodological characteristics of interest? The concept of methodological characteristics should be further developed.

This point is well taken. Our intention overall is to review strategies and approaches that have been used to develop, test, or implement EM screenings or interventions for persons with cognitive impairments or their family members. To your point, yes, the methodological characteristics described on p.10 under Charting the data are the aspects we are interested in extracting to accomplish our larger goal. To help clarify in the manuscript, we have adjusted the language in the Charting the data section (first sentence) as follows:

“Covidence software will be used to extract data on relevant methodological characteristics from each included peer-reviewed article that comprise overall approaches used to develop, test, or implement EM screenings or interventions for older adults in the context of cognitive impairment.”

We have also tightened up the language in the introduction to help emphasize our broader intention:

Starting line 114: “The purpose of this proposed scoping review is to identify, summarize, and compare the unique methodological considerations and practical approaches that have been used to develop and implement EM screening tools or interventions for individuals with cognitive impairment or their family members.”

2. Sub-question 1 identifies design modifications – is this different from characteristics? Also, modified from what? Will there be comparisons between studies that examine EM in older people in general and older people with cognitive impairment?

We appreciate this point. We did intend “design modifications” as a characteristic we would extract from our collected data. By modifications, our intent is to examine practical choices noted by researchers to adapt their study design or approaches (e.g., recruitment, methods, etc.) to benefit the development, testing, or application an EM screener or intervention in the context of persons with cognitive impairment. To help clarify, in the Charting the data section when listing the characteristics that we would extract, we made an edit reflecting this meaning of modification. (starting line 226): “…and authors’ conclusions or noted implications for considering cognition and making research design modifications for this population.

To address the last point, we do not intend compare studies examining EM in older people with cognitive impairment vs. in general, in that our intent is to focus on methodological considerations for the former group.

3. It is not clear if a similar scoping review/systematic review already exists on the topic. This should be identified.

Thank you for this comment. We have brought more attention to the one prior review that we found that evaluates studies assessing EM among older adults with dementia and juxtaposed this study with ours. Starting line 108: “Moreover, while a prior review has conveyed how variation in EM definitions and measurements impact EM findings for older adults with dementia compared to those without [12], less is known about the practical considerations and modifications adopted by researchers when developing and implementing EM assessments and intervention strategies for individuals with varying levels of cognitive impairment, many of whom may be unable to fully communicate their experiences of abuse.”

4. Line 141 states that a quality appraisal will not be done. Given this is a scoping review, this is not necessary to state.

We have deleted this sentence.

5. Given the research question/sub-questions identified, it is not clear how the findings from this review will identify the challenges in identifying the prevalence of EM (page 6 127-128).

Thank you for this comment. We have worked to clarify with the following language.

Regarding the section on pg 6 (starting line 119), we made the following change: “In using this approach to map out existing literature on EM assessment and intervention strategies for persons with varying levels of cognitive impairment, this review will inform future studies by summarizing previously identified challenges and recommendations for implementing EM screenings and interventions to people experiencing impaired cognition and their caregivers.”

We also made adjustments in the abstract (starting line 36): “Through mapping out existing strategies and approaches used to develop, test, or implement EM screening tools or interventions, this review will outline previously identified recommendations and challenges pertinent to future EM assessment, reduction, and prevention efforts.”

Finally, we adjusted language in the discussion (starting second sentence): “By summarizing effective tactics, challenges, and research gaps that have been previously reported in relation to EM assessment among this population, findings from this review will also inform efforts to develop EM screening tools and interventions specifically for these individuals.”

6. Like the above comment, given this is a scoping review with no critical appraisal, it will not be possible to offer recommendations as stated in line 128.

We have worked to clarify this point also through the edits described above in comment 5.

7. Type of outcome – it is possible that a study could meet the inclusion criteria by reporting signs and symptoms of abuse but not identify a specific assessment tool. It is not clear if these studies would be included.

We appreciate this point. We have added to our exclusion criteria (#2.) to indicate that we will exclude “studies that do not incorporate an assessment tool to determine the presence of EM or EM risk.”

8. It is interesting that nothing has been discussed about the severity of cognitive impairment (e.g. MMSE) it only states “varying degrees”.

In our protocol, we have used the phrasing “varying degrees of cognitive impairment,” and in a few places “older adults with cognitive impairment of varying severity,” to convey our intent to review and report on studies that have assessed EM among individuals experiencing any identified level of cognitive impairment. This could be, for example, individuals with mild cognitive impairment or those having a more severe diagnosis. Within this focus, we intend to report on different study methods that have been used to determine patients’ cognitive impairment status (e.g., using MMSE) though we are not considering severity as a type of inclusion criteria.

Reviewer #2: Thank you for the opportunity to review your manuscript. Your objective, justification, research questions, and inclusion criteria align nicely. Your topic is incredibly important and as a witness of various forms of elder mistreatment, I am so pleased to see your research plans!

Thank you for this feedback!

My comments are very minor and are aimed to bring more clarity to your concepts to enhance rigor. Overall, I enjoyed this manuscript and found it very thorough and well done.

1. Line 65: you can put just “EM” instead of elder mistreatment

Thank you for catching this! We have made the correction. (Line 58)

2. Line 127-128: “… this review will inform future research through identifying challenges and recommendations for developing and implementing EM screenings and interventions for individuals with cognitive impairment.” I suggest changing this wording – scoping reviews are not meant to identify recommendations, but they can map what recommendations have been made already.

Thank you for this comment. We have adjusted the wording to reflect that this review will: “inform future studies through summarizing previously identified challenges and recommendations for implementing EM screenings and interventions to people with impaired cognition and their caregivers.” (Starting line 120)

3. Line 126, 188: Prevention strategies – these are the only two places prevention/mitigation strategies are mentioned. It seems to me this may not align well with your purpose/questions – perhaps reword or reconsider this for clarity.

We used the term prevention in the manuscript just as studies we are considering can include those developing or implementing an intervention testing preventative strategies for EM. To your point and to add consistency to the protocol, starting on line 37, we have edited the sentence as: “…this review will outline previously identified recommendations and challenges pertinent to future EM assessment, reduction and prevention efforts.” We have also changed “prevention” to “intervention” on line 119 and on line 190, we cut the term “prevention” and mitigate”.

4. Lines 155-158: Perhaps include why these databases were chosen.

Absolutely. We have added a rationale for these databases on pg. 7 (starting line 150). It reads:

These databases were selected based on availability of the resources and relevancy of the content covered. Medline is the premier biomedical database, CINAHL covers other health sciences fields, Cochrane Trials provides evidence-based literature, and Scopus provides a large and multidisciplinary scope of the scholarly literature.

5. Line 170: Covidence is fantastic for scoping reviews! However, your readers may not know what Covidence is. I suggest quickly adding something about Covidence being a software used to streamline the knowledge synthesis process

Thank you for this recommendation to clarify our use of Covidence. We have adapted this sentence (starting line 164) to read: “Moving forward, search results will be extracted and imported into Covidence, a web-based tool used to streamline the process of knowledge synthesis for reviews, and all duplicates will be removed.”

6. Line 172: Similar to the previous comment, what is the SDMO framework and why is using it important for the screening process where you are already following Arksey and O’Malley’s framework and using Covidence. How does SDMO come into play?

This comment is helpful in clarifying our reasoning for using the Study Design, Data, Methods, Outcomes (SDMO) framework. We have added sentence (starting line 183) describing its usefulness for determining inclusion criteria for methodologically focused reviews: “This framework has been recommended for methodologically focused reviews, as it guides identification of study design, data, and outcome elements relative to the research question [38].”

7. Line 185: It is nice to have very clear conceptual definitions in inclusion criteria for scoping reviews. Although you defined EM in the introduction, consider giving a clear definition of what you will consider as EM in your review within the inclusion criteria. For example, “EM includes but is not limited to any intentional or unintentional emotional, physical, financial, or sexu

---

## [Editor Report · Decision Letter 1]

23 Feb 2025

Methodological considerations for assessing elder mistreatment of older adults with cognitive impairment: A scoping review protocol

PONE-D-24-36659R1

Dear Dr. Schlag,

We’re pleased to inform you that your manuscript has been judged scientifically suitable for publication and will be formally accepted for publication once it meets all outstanding technical requirements.

Kind regards,

Patricia Anne Morris

Academic Editor

PLOS ONE

Additional Editor Comments (optional):

Your response to reviewers was thorough and the revisions are well done.
---

## [Editor Report · Acceptance letter]

PONE-D-24-36659R1

PLOS ONE

Dear Dr. Schlag,

I'm pleased to inform you that your manuscript has been deemed suitable for publication in PLOS ONE. Congratulations! Your manuscript is now being handed over to our production team.

Kind regards,

on behalf of

Dr. Patricia Anne Morris

Academic Editor

PLOS ONE